# Photoswitchable Zirconium MOF for Light-Driven Hydrogen Storage

**DOI:** 10.3390/polym13224052

**Published:** 2021-11-22

**Authors:** Vera V. Butova, Olga A. Burachevskaya, Vitaly A. Podshibyakin, Evgenii N. Shepelenko, Andrei A. Tereshchenko, Svetlana O. Shapovalova, Oleg I. Il’in, Vladimir A. Bren’, Alexander V. Soldatov

**Affiliations:** 1The Smart Materials Research Institute, Southern Federal University, Sladkova 178/24, 344090 Rostov-on-Don, Russia; oburachevskaya@sfedu.ru (O.A.B.); antereshenko@sfedu.ru (A.A.T.); scherkasova@sfedu.ru (S.O.S.); soldatov@sfedu.ru (A.V.S.); 2Institute of Physical and Organic Chemistry, Southern Federal University, 344090 Rostov-on-Don, Russia; vpodshibakin@sfedu.ru (V.A.P.); vabren@sfedu.ru (V.A.B.); 3Federal Research Center the Southern Scientific Center of the Russian Academy of Sciences, 344006 Rostov-on-Don, Russia; e-shepelenko@mail.ru; 4Institute of Nanotechnologies, Electronics and Equipment Engineering, Southern Federal University, Shevchenko 2, 347922 Taganrog, Russia; oiilin@sfedu.ru

**Keywords:** UiO-66, smart material, diarylethene, UiO-66-NH_2_, photochromism, activation, FTIR, in situ IR-spectroscopy

## Abstract

Here, we report a new photosensitive metal–organic framework (MOF) that was constructed via the modification of UiO-66-NH_2_ with diarylethene molecules (DAE, 4-(5-Methoxy-1,2-dimethyl-1H-indol-3-yl)-3-(2,5-dimethylthiophen-3-yl)-4-furan-2,5-dione). The material that was obtained was a highly crystalline porous compound. The photoresponse of the modified MOF was observed via UV–Vis and IR spectroscopy. Most of the DAE molecules inside of the UiO-66-pores had an open conformation after synthesis. However, the equilibrium was able to be shifted further toward an open conformation using visible light irradiation with a wavelength of 520 nm. Conversely, UV-light with a wavelength of 450 nm initiated the transformation of the photoresponsive moieties inside of the pores to a closed modification. We have shown that this transformation could be used to stimulate hydrogen adsorption–desorption processes. Specifically, visible light irradiation increased the H_2_ capacity of modified MOF, while UV-light decreased it. A similar hybrid material with DAE moieties in the UiO-66 scaffold was applied for hydrogen storage for the first time. Additionally, the obtained results are promising for smart H_2_ storage that is able to be managed via light stimuli.

## 1. Introduction

Metal–organic frameworks (MOFs) are porous materials that have recently attracted a great deal of attention do their high-potential for use in molecule design [1,2]. These materials are constructed from inorganic clusters and organic molecules. The first ones tended to be designated as secondary building units (SBUs) [3]. They often contained one metal ion or a group of ions coordinated by oxygen or nitrogen from organic components, which are also known as linkers. Both SBUs and linkers define the geometry and functionality of the resulting framework. The introduction of functional groups via linker molecules results in the pore surface being decorated with them. Because of this, they are able to manipulate the properties of a material in the desired way. As a result, MOFs have been successfully applied in fields such as gas storage and separation [4,5], biomedicine [6,7], environmental remediation [8,9,10], and catalysis [11,12]. 

For modern devices, it is essential to control the material properties via external stimuli. Light is to be considered a preferred type of stimuli because it can be controlled precisely in terms of both wavelength and radiation time [13]. In this way, imparting photoresponsive moieties to MOFs allows their fundamental properties to be managed. Different strategies have been applied to the design of new generations of photoswitchable MOFs. Photochromic molecules of a different nature have been introduced to MOFs [14]. Among other photochromic compounds, diarylethenes (DAEs) are one of the most promising due to their fatigue-resistant and thermally irreversible performance [15]. These molecules are able to reverse their configuration from that of a colorless open isomer to a colored closed form under UV-light irradiation. During this transformation, chemical bonds are formed between two heterocyclic thiophene groups. Visible light initiates a reverse process [16]. Generally, DAE molecules can be incorporated into the MOF scaffold in two ways: as parts of linker molecules [17,18,19] or as guests in the pores [20,21]. 

The photochromic process in DAE molecules affects the available pore volume, so it was applied for control adsorption and release of CO_2_ [22,23,24,25] and gas separation [26,27,28] in previous studies. DAE molecules were introduced to MOFs to control the energy transfer process. Shustova and coauthors reported Zn-MOF with DAE and porphyrin species [29]. In this report, DAE was used to manage the fluorescence of porphyrin. Later Zhou and co-workers proposed a similar system to control the generation of singlet oxygen [30]. 

A list of MOFs was applied for targeted functionalization. One of the most popular MOFs for this purpose is UiO-66 and its derivates. The main reason for its popularity is its stability and high porosity. UiO-66 is constructed from zirconium ions and terephthalate linkers. Each SBU in UiO-66 contains six Zr^4+^ ions and bridge oxygen atoms and is able to be represented with a Zr_6_O_4_(OH)_4_ formula unit (Figure 1). After the activation procedure, the bridge µ_3_-OH-group loses H^+^, and the SBU formula of the activated sample can be assigned to Zr_6_O_6_. This equilibrium process leads to UiO-66 and its derivatives presenting acidic properties [7,31,32]. Each SBU is coordinated with 12 linker molecules. High connectivity and strong covalent bonds result in the high stability of the UiO-66 scaffold. The introduction of functional groups into the terephthalate linker results in the functionalization of the pore surface, and it is repeatedly applied to manipulate the properties of MOF for use in desired applications [4,33,34]. 

Here, we report the synthesis procedure for the incorporation of DAE-molecules to the UiO-66-NH_2_ framework. We used covalent the bonding of photoresponsive moieties and porous scaffolds to avoid the leakage of functional species. 

## 2. Materials and Methods

### 2.1. Synthesis

The starting materials zirconium tetrachloride (ZrCl_4_), amino terephthalic acid (NH_2_-BDC), benzoic acid (BA), N, N-dimethylformamide (DMF), and isopropanol were purchased from commercial suppliers and were used without further purification. Deionized water was obtained from the the Simplicity UV water purification system (Merck Millipore). 

UiO-66-NH_2_ was synthesized according to a previously reported technique [7]. Briefly, ZrCl_4_, NH_2_-BDC, deionized water, and BA were completely dissolved in DMF. The molar ratio ZrCl_4_: NH_2_-BDC: H_2_O: BA: DMF was 1: 1: 3: 10: 300. The reaction mixture was placed into a preheated oven and was held at 120 °C for 24 h. After being cooled, the precipitate was collected via centrifugation and was washed twice with DMF and with isopropanol. It was dried at 60 °C for 12 h. 

The compound 4-(5-Methoxy-1,2-dimethyl-1H-indol-3-yl)-3-(2,5-dimethylthiophen-3-yl)-4-furan-2,5-dione (DAE) was obtained according to previously published procedures [35].

The modification of UiO-66-NH_2_ was performed according to the following scheme (Figure 2). 

The modification of UiO-66-NH_2_ was performed according to previously reported methods, with some modifications that were made based on the specific objective of obtaining diarylethenes with pyrrole-2,5-dione bridging fragments [35,36,37]. A solution containing 0.36 mmol of DAE in isopropyl alcohol (30 mL) was treated with UiO-66-NH_2_ (0.06 mmol) and 4-dimethylaminopyridine (DMAP) (1 mg). The reaction mixture was refluxed for 10 h and cooled, and the solvent was removed by means of distillation. The resulting precipitate was washed with CH_2_Cl_2_ until no staining was detected. The obtained sample was designated as DAE-UiO-66. 

### 2.2. Methods

Powder X-ray diffraction (XRD) profiles were collected on an X-ray diffractometer Bruker D2 Phaser (CuKα, λ = 1.5417 Å) with a scanning rate of 0.2 s/step and a step of 0.01°. Profile analysis was performed in Jana2006 software [38]. For X-ray fluorescence (XRF) analysis, we used the Bruker M4 Tornado. Thermogravimetric analysis (TGA) and differential scanning calorimetry (DSC) were performed using STA 449 F5 Jupiter in airflow. 

A Nova Nanolab 600 scanning electron microscope (SEM) (FEI, Netherlands) was used for sample topological control. A Genesis SPECTRUM spectrometer (EDAX AMETEK, USA) was used for the elemental composition analysis, which was achived through energy-dispersive X-ray spectroscopy (EDX). The EDX spectra were obtained with a 15 kV accelerating voltage and an 8.4 nA electron beam current. The signal accumulation time was 100 s.

N_2_ and H_2_ adsorption–desorption isotherms were collected using an ASAP 2020 Accelerated Surface Area and Porosimetry analyzer (Micromeritics) at −196 °C. The samples were activated under a dynamic vacuum at 90 °C for 12 h before measurements were taken. To calculate the specific surface areas, we used N2-adsorption data and the BET model [39]. Pore size distribution is presented according to Non-Local Density Functional Theory (see details in Appendix A). For the experiments with light-irradiation, we used two types of lasers—a green laser with a wavelength of 520 nm to trace the visible-light effects and a UV-laser with a wavelength 450 nm for the evaluation of the UV-light effects. The sample in the glass vessel was irradiated using the green laser and by emitting it through the glass for 10 min. We moved the laser point to avoid the excessive heating of the sample. The vessel was opened for irradiation with UV light, and the sample was treated with the UV light directly for 10 min. After this, the sample was closed and degassed at 50 °C for 2 h in a dynamic vacuum. 

IR spectra were measured on a Bruker Vertex 70 spectrometer in DRIFT (Diffuse reflectance) geometry using an MCT detector and a Praying Mantis Low-Temperature Reaction Chamber (Harrick) attachment. The spectra were measured in the range from 5000 to 500 cm^−1^ and had a resolution of 1 cm^−1^ and scanning for 1 min for each spectrum. The reference sample was KBr powder. For sample activation, we used outgassing at 90 °C for 30 min and we then cooled the sample down to room temperature with further outgassing. After activating and recording initial spectra, both samples were irradiated with a green laser (520 nm) for 2 min, and the spectra were recorded under these stimuli. 

UV–Vis spectra were collected on a UV-2600 (Shimadzu) spectrophotometer with an integrating sphere. The samples (14 mg) were mixed with BaSO4 (300 mg), ground, mixed and pressed in the substrate vessel. After recording the initial spectra, samples were irradiated under a Newport lamp (power 500 W) with a filter passing 436 nm light for 10 min. After this, we recorded the spectra that were obtained from the treated samples. 

## 3. Results

According to the XRD data, the UiO-66-NH_2_ sample did not demonstrate a different crystal structure type after functionalization (Figure 3a). The SEM images also show that modification with the DAE-molecules did not affect the morphology of the particles (Figure 3b). The DAE-UiO-66 sample consisted of octahedral crystals that were 60–90 nm in size (Appendix A). EDX-mapping of the DAE-UiO-66 sample revealed the uniform distribution of Zr, C, N, and S atoms (Appendix A). Both of the UiO-66-NH2 and DAE-UiO-66 samples had cubic symmetry with the Fm-3m (225) space group. However, profile analysis revealed that there was an increase in lattice constants, which increased from 20.794 Å (UiO-66-NH_2_) to 20.836 Å (DAE-UiO-66) (Appendix A). This could indicate stress in the framework of the DAE-UiO-66 sample due to the pores becoming filled with relatively photochromic molecules. 

In good agreement with the XRD-data, we observed that the DAE-UiO-66 sample demonstrates lower porosity compared to the initial sample (Table 1). Figure 4a represents the nitrogen sorption isotherms of the UiO-66-NH_2_ and DAE-UiO-66 samples. The shapes of both of the isotherms are associated with the microporous nature of the samples and can be assigned to type I as per IUPAC notification. Both isotherms demonstrate pronounced steps in the low-pressure region and a subsequent plateau due to the formation of a N_2_ monolayer in the micropores. At relative pressures above 0.8, hysteresis loops of the H1 type could be observed, which is associated with the capillary condensation of nitrogen in cylindrical mesopores. The crystal structure of UiO-66-NH_2_ contains two kinds of pores—tetrahedral and octahedral ones. The pore size distribution for the UiO-66-NH_2_ and DAE-UiO-66 samples was calculated using the DFT method according to cylindrical pores (Appendix A). It was shown that the small tetrahedral pores that were present on DAE-UiO-66 sample did not change in size compared to those found on the UiO-66-NH_2_ sample, but the available volume of such pores was significantly reduced. While octahedral pores were present on the DAE-UiO-66 sample, their diameter was reduced. We suppose that this could indicate that photochromic molecules are located in octahedral pores, reducing their diameter. Tetrahedral pores are too small for such molecules, so they are empty, but most of them are blocked by photochromic molecules in the neighboring octahedral pores. 

Figure 4b represents the TGA and DSC curves for the UiO-66-NH_2_ and DAE-UiO-66 samples. The TGA curves of both samples show two stages. The first weight loss occurs in the temperature range of 25–110 °C and is the result of the evacuation of water molecules from the porous MOFs. This process is endothermic and corresponds to negative packs in the DSC curves. For the UiO-66-NH_2_ sample, this step is more pronounced due to the higher porosity of the material before modification. The weight loss that occurs in the temperature range of 300–550 °C is associated with the decomposition of the organic parts of frameworks and the resulting structural collapse. The TGA curves were normalized according to the amount of solid residual waste that remained after annealing the samples in the airflow. The interaction of both samples with oxygen from the air leads to the formation of six units of ZrO_2_ from each formula unit of the respective sample as the only solid residual. All organic components (linkers for the UiO-66-NH_2_ sample and linkers and DAE-moieties for the DAE-UiO-66 sample) with oxygen form gaseous products. As such, the solid residual and weight loss of the samples could be recalculated in the composition of the materials (see Appendix A). Additional DAE-molecules in the UiO-66-scaffold led to more significant weight loss. Specifically, we calculated that one of each of the ten linkers in the DAE-UiO-66 sample was modified with the DAE-molecules (Table 1). 

Additionally, we applied XRF analysis to quantify the amount of DAE molecules in the DAE-UiO-66. The unmodified UiO-66-NH_2_ sample contains Zr and Cl in a molar ratio of 6:3. As previously reported, certain amino groups are able to interact with Cl- ions during synthesis due to the high acidity of the reaction mixture [40]. According to the XRF data, the formula unit of UiO-66-NH_2_ could be represented as Zr_6_O_4_(OH)_4_(C_8_H_5_O_4_N)_3_(C_8_H_6_O_4_NCl)_3_. The DAE-UiO-66 sample contains Zr, Cl, and S in a molar ratio of 6:0.3:0.5 (Table 1). Thus, the formula unit of DAE-UiO-66 is Zr_6_O_4_(OH)_4_(C_8_H_5_O_4_N)_5_._2_(C_8_H_6_O_4_NCl)_0_._3_(C_29_H_23_O_7_N_3_S)_0_._5_. One octahedral pore is constructed from 12 linker molecules with 12 amino groups. According to the XRF data, one octahedral pore contains one DAE molecule. Moreover, we observed a reduction in the concentration of the -NH_3_Cl-moieties in the DAE-UiO-66 sample. This could indicate the formation of intermolecular bonds between DAE and the neighboring amino groups. 

We traced the response of the synthesized compounds on the green laser with a wavelength 520 nm using FTIR spectroscopy (Figure 5). In the fingerprint region, both spectra are dominated by bands of the UiO-66-NH_2_ framework. The only difference is a new mode at 1000 cm^−1^ on the spectrum of the DAE-UiO-66 sample, which can be attributed to the C=C vibrations [41]. The as-synthesized samples contain a lot of adsorbed water molecules, which obstruct the interpretation of the far-infrared region of the spectrum. We evacuated the adsorbed water before conducting the measurements (Appendix A). Three specific bands were observed for the DAE-UiO-66 sample: at 3065, 2925, and 2853 cm^−1^, which are designated as (7), (8), and (9) in Figure 4b, respectively. We attributed them to the vibrations of the C-H bonds. The spectrum of the un-modified UiO-66-NH_2_ sample contains a list of bands in this region, which are associated with the C-H vibrations of the linkers [7]. The amino group vibrations result in increases being present in the bands at at 3510 and 3400 cm^−1^ (lines (2) and (3) in Figure 5b) in the UiO-66-NH_2_ spectrum [7,31]. The modification of the sample results in a shift of these modes to 3490 and 3370 cm^−1^, respectively. We attribute this to the involvement of this NH_2_-groups in the formation of bonds during modification. For activated samples, the modification effect on photo-response was traced. The UiO-66-NH_2_ sample did not exhibit any significant response on laser irradiation, while the spectrum of the DAE-UiO-66 sample after irradiation contained a list of changes. The most pronounced one was an almost complete disappearance of band at 3670 cm^−1^ (designated as (1) in Figure 5b). This band can be attributed to the vibrations of the µ_3_-OH groups in the UiO-66 framework. The deprotonation of such species results in a reduction in its intensity [31,34]. The band at 3330 cm^−1^ in the UiO-66-NH_2_ spectrum before irradiation refers to H-bonded water molecules [31,34]. After modification, the DAE-UiO-66 spectrum demonstrates the shift of this band to 3280 cm^−1^ due to the formation of hydrogen bonds with more electronegative species [31]. We suppose that O=C-groups could form such bonds in a modified sample. After irradiation with a laser, the photochromic process affects the orientation of the molecules with the breaking of these bonds. In good agreement with this, we observed that the corresponding band vanished in the spectrum of the irradiated DAE-UiO-66.

Figure 6a represents the UV–Vis spectra of the UiO-66-NH_2_ and DAE-UiO-66 samples. After modification with the DAE molecules, the UiO-66-NH_2_ spectrum preserved all of the characteristic features observed at 240, 265, and 375 nm. The introduction of DAE molecules resulted in three new modes at 220, 295, and 450 nm. This indicates that the DAE molecules were incorporated into the UiO-66-NH_2_ structure in the open form [30]. The irradiation of the DAE-UiO-66 with UV-light gave rise to a new band at 545 nm (Figure 6b), which is in good agreement with the formation of a closed DAE-modification. 

We have traced the hydrogen capacity for the DAE-UiO-66 sample that was consistently irradiated with visible and UV light. The isotherms are presented in Figure 7a. After irradiation with visible light (wavelength 520 nm), the as-synthesized sample adsorbed 0.58 weight% of hydrogen at 77 K. This value is slightly higher than the H_2_ capacity for the as-synthesized sample (0.55 weight%). Irradiation with UV-light (450 nm) reduced the hydrogen capacity. These changes are reversible because the same trend was reproduced after further irradiations with visible and UV light (Figure 7b). According to obtained data, we were able to conclude that the closed modification of the DAE molecules results in a lower pore volume being available than in an open modification. The same trend was reported for CO_2_ adsorption [42] and C_2_H_2_/C_2_H_4_ separation [27]. In order to trace the possible structural degradation after light stimulation, we measured the SSA for samples after each step. After the second irradiation with visible light, the SSA of the DAE-UiO-66 sample was estimated as 500 m^2^/g, indicating the porous structure of the framework without degradation (Appendix A). We also did not observe any significant effect of light irradiation on the XRD patterns of the DAE-UiO-66 sample, indicating that transformations inside of the pores did not result in a structural collapse (Appendix A). 

## 4. Discussion

According to the experimental data, we propose that the DAE molecules undergo the following transformations inside of the pores (Figure 8): After the synthesis, most of the DAE molecules are in the “open” modification, as confirmed by UV–Vis spectra. After irradiation with visible light, this equilibrium strongly shifts to the left, resulting in the open form being the dominant DAE modification. The open form increased hydrogen capacity and resulted in pronounced changes in the IR spectrum. According to the last experiment, irradiation with light (520 nm) led to a proton transfer from the μ_3_-OH group of the SBU in DAE-UiO-66. We attribute this transfer to the presence of DAE molecules inside of the pores because it was not observed in the non-modified UiO-66-NH2 sample. It has been reported that the DAE molecules cannot be protonated in an acidic medium [43,44]. We suppose that the C=O groups of the DAE molecules could coordinate the H+ ions. This transfer was promoted due to open conformation, allowing the μ_3_-OH groups to be close to the SBU and C=O groups of the DAE molecules. Under UV-irradiation, the DAE molecules underwent a “closed” modification, which was determined according to UV–Vis spectroscopy. Additionally, this resulted in the DAE-UiO-66 sample having a lower H_2_ capacity. 

## 5. Conclusions

In summary, we modified UiO-66-NH_2_ with the photoactive DAE molecule 4-(5-Methoxy-1,2-dimethyl-1H-indol-3-yl)-3-(2,5-dimethylthiophen-3-yl)-4-furan-2,5-dione. These DAE molecules were used to modify the MOF scaffold for the first time. Moreover, the DAE moieties were incorporated into the linker of the canonical MOF, UiO-66NH_2_. The obtained material was highly crystalline and was isostructural to the initial UiO-66-NH_2_ scaffold. However, we noticed an increase in the lattice constants, which was caused by stress due to the molecules in the pores. However, the obtained material, DAE-UiO-66, demonstrated a preserved porous structure, and its SSA was estimated to be almost 400 m^2^/g. According to a complex analysis of the IR-spectra, UV–Vis spectra, and hydrogen adsorption isotherms, we can conclude that the as-modified sample contained a mixture of two forms of DAE-molecules—”open” and “closed”. The equilibrium was shifted to the “open” modification from the very beginning. However, some changes could be observed after the irradiation of DAE-UiO-66 with a visible light source, initiating the transformation of the DAE fillers in the pores to the open conformation. This increased the hydrogen capacity. Secondly, we observed the transport of protons from the μ_3_-OH groups of UiO-SBUs to the C=O groups of the DAE-molecules. Such a phenomenon was only able to be observed after the activation of MOF in dry air. The open–close transformations were also confirmed via UV–Vis spectroscopy. Finally, we measured the hydrogen capacity under UV/Vis irradiation and observed a reversible process that was able to be managed via light stimuli. Specifically, visible light increases the H2 capacity, while UV-irradiation decreases it. We suppose that this result is an important step in the development of smart hydrogen storage with light-induced safe H_2_-desorption. 

## Figures and Tables

**Figure 1 polymers-13-04052-f001:**
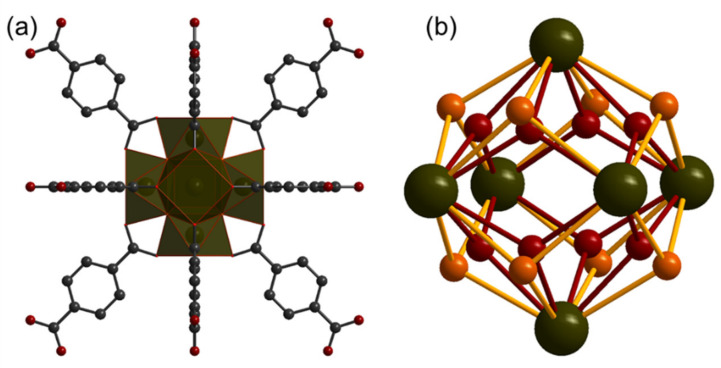
Schematic representation of the UiO-66 structure according to crystallographic data COD ID 4512072. (**a**) Dark green polyhedral represents the Zr-O cluster; gray spheres represent carbon atoms, dark red represents for oxygen. (**b**) Schematic representation of Zr_6_O_4_(OH)_4_ SBU. Both oxygen positions are provided: µ_3_-O (dark red) and µ_3_-OH (orange). Oxygen atoms can alternate to occupy each of them The picture was prepared in Diamond, version 4.6.5.

**Figure 2 polymers-13-04052-f002:**
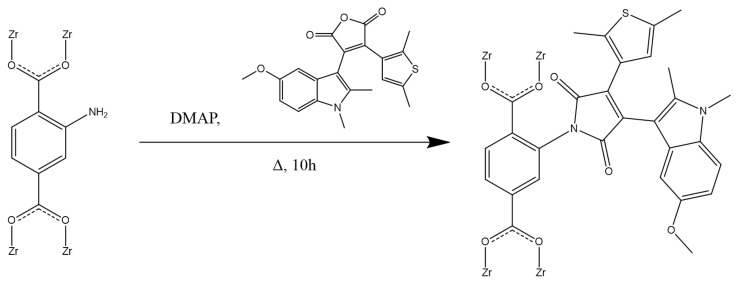
Scheme of UiO-66-NH_2_ functionalization.

**Figure 3 polymers-13-04052-f003:**
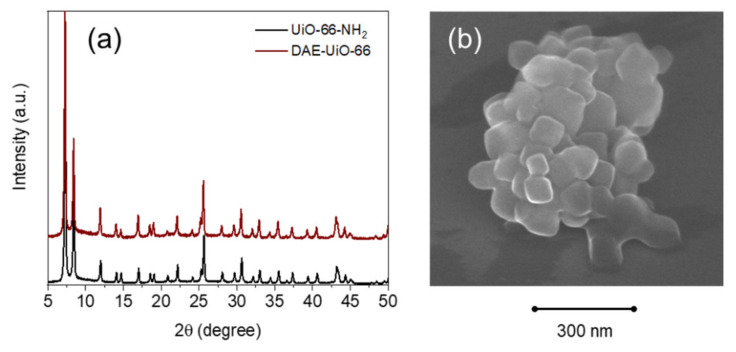
(**a**) XRD profiles of the UiO-66-NH_2_ (black) and DAE-UiO-66 (brown) samples. Intensities are normalized and shifted along the *y*-axis for better representation. (**b**) SEM image of DAR-UiO-66 sample.

**Figure 4 polymers-13-04052-f004:**
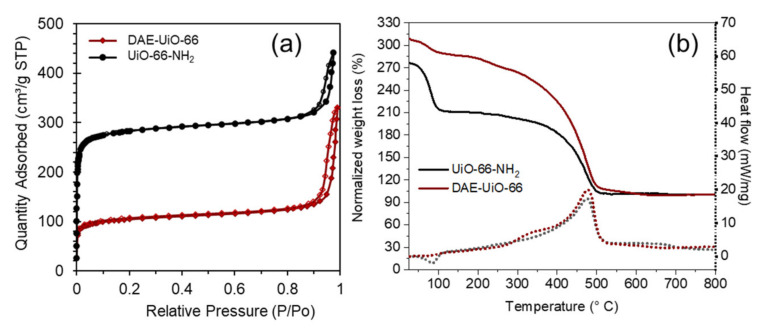
(**a**) Nitrogen adsorption–desorption isotherms of the UiO-66-NH2 (black) and DAE-UiO-66 (brown) samples. Filled markers designate isotherm adsorption branches; empty markers represent desorption branches. (**b**) TGA (solid lines) and DSC (dashed lines) curves of UiO-66-NH_2_ (black) and DAE-UiO-66 (brown) samples. TGA curve was normalized according to solid residual (see details in Appendix A).

**Figure 5 polymers-13-04052-f005:**
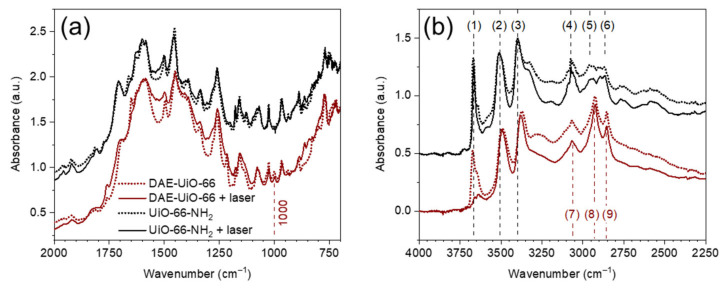
FTIR spectra of UiO-66-NH_2_ (black ones) and DAE-UiO-66 (brown ones) samples in the 700–2000 cm^−1^ (**a**) and 2250–4000 cm^−1^ (**b**) region. In both parts, spectra of the samples before laser irradiation are represented by dotted lines, while solid lines show spectra of the samples after 10 min of irradiation (wavelength 520 nm).

**Figure 6 polymers-13-04052-f006:**
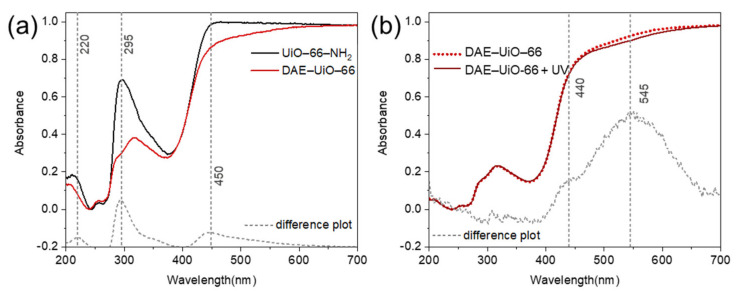
(**a**) UV–Vis spectra of samples UiO-66-NH_2_ (black curve) and DAE-UiO-66 (red curve). (**b**) UV–Vis spectra of DAE-UiO-66 sample (dotted curve) and those after irradiation with light (436 nm) for 10 min (solid line). For both parts, difference plots are provided as dashed gray lines at the bottom.

**Figure 7 polymers-13-04052-f007:**
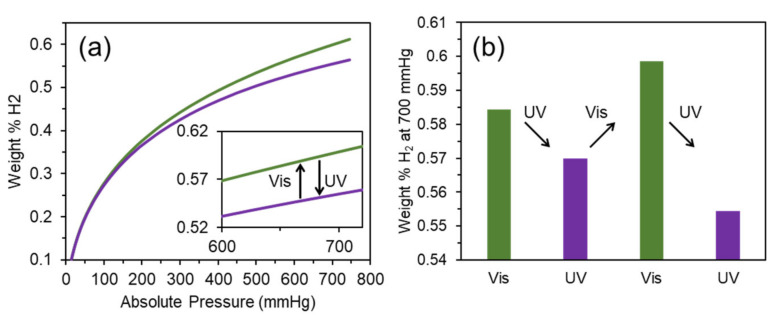
(**a**) Isotherms of hydrogen adsorption (77 K) for DAE-UiO-66 sample irradiated with visible light with a wavelength of 520 nm (green plot) and UV-light with a wavelength of 450 nm (violet plot). (**b**) Diagram of changing in H_2_ capacity during sequential irradiation with visible light with a wavelength of 520 nm (green columns) and UV-light with a wavelength of 450 nm (violet columns).

**Figure 8 polymers-13-04052-f008:**
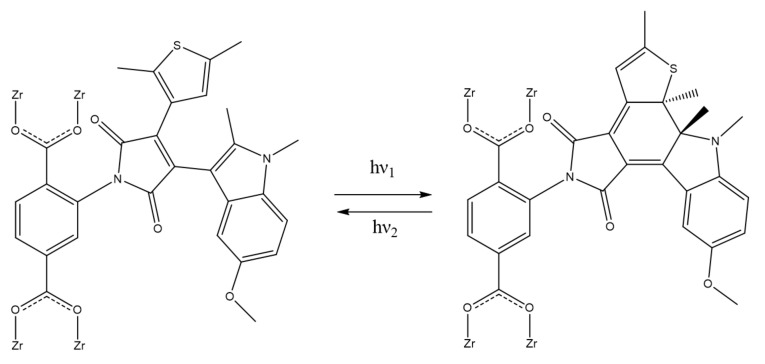
Scheme of DAE transformations inside DAE-UiO-66 pores under UV-light (hν_1_) and visible light (hν_2_).

**Table 1 polymers-13-04052-t001:** Some properties of the synthesized samples UiO-66-NH_2_ and DAE-UiO-66. SSA stands for the specific surface area. DAE content was calculated in mol %.

Samples’ Designation	Unit Cell Parameters		Nitrogen Adsorption	TGA			XRF	
	a, Å	V, Å^3^	SSA, m^2^/g	Weight Loss, %	Molecular Weight	DAE Content	Zr:S:Cl	**DAE Content**
UiO-66-NH_2_	20.794(5)	8991(4)	1111	51.3	1519	-	6:0:3	-
DAE-UiO-66	20.836(6)	9046(5)	398	62.4	1945	9%	6:0.5:0.3	8%

## Data Availability

Data is contained within the article and Appendix A.

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
