# Peer review of "Photoswitchable Zirconium MOF for Light-Driven Hydrogen Storage"

_polymers, 2021, doi:10.3390/polym13224052_

Round 1

Reviewer 1 Report

This article was reviewed by me in its previous version. It must be admitted that the Authors responded to most of my comments and modified the work, especially in terms of supplementing the information that raised concerns about copyright. At this point, I have no further comments.

However, I still have substantive comments:

– In many places the authors do not use specialized vocabulary for a given method. For example, in spectroscopy we are talking about bands, not peaks – please fix that.

– In my opinion, the sorption isotherm with a distinct hysteresis loop is type IV characteristic of mesoporous materials.

– d spacing should be given for each peak in the diffraction pattern.

The above observations do not allow the reviewer to recommend the paper in its current form for publication, however I can recommend its publication after minor revisions.

Author Response

The authors appreciate carefully reviewing the manuscript. 

– In many places the authors do not use specialized vocabulary for a given method. For example, in spectroscopy we are talking about bands, not peaks – please fix that.

Thank you for the comment. "Peaks" in the text were replaced with "bands". 

– In my opinion, the sorption isotherm with a distinct hysteresis loop is type IV characteristic of mesoporous materials.

I totally agree with the fact, that the hysteresis loop indicates mesopores. However, in this case, we consider a complex system. Sample UiO-66NH2 consists of porous nanoparticles. The pores crystals are about 1 nm, according to crystal structure. So at low pressures, we observe the fast formation of N2 monolayer, which is illustrated by pronounced step with a subsequent plateau in N2 adsorption isotherm. Up to relative pressure p/p0 = 0.8, the isotherm shape is a typical microporous type I. At higher pressures, we observe a hysteresis loop due to capillary condensation of N2 in mesopores.  These pores are formed between uniform nanoparticles in aggregates. So instead of mesoporous material, we have microporous nanoparticles, which interparticle cavities in aggregates imitate mesopores. The same situation was previously observed by us (10.1021/acs.inorgchem.0c03751), and by other authors (10.1039/C8DT03312A , 10.1021/acsomega.0c00687).

– d spacing should be given for each peak in the diffraction pattern.

d-spacing is given now in Table S2 in SI. 

Reviewer 2 Report

The authors' responses to the reviews are satisfactory. 

Author Response

The authors are grateful for the positive feedback. 

This manuscript is a resubmission of an earlier submission. The following is a list of the peer review reports and author responses from that submission.

Round 1

Reviewer 1 Report

This manuscript reports of research on the modification of synthetic UiO-66-NH2. Review of related literature is quite well developed and provides a good introduction to the experimental part. Figures are legible and carefully prepared.

I would recommend following points to the attention of the authors:

– Where did the parameters for the modification of the UiO-66-NH2 structure come from? Have the authors conducted comprehensive research or is it rather a single hit? In my opinion, the work that I would classify as basic research should include the entire range of the concentrations used, especially since the authors do not provide a reference to their previous papers.

– Line 67 – Did the authors prepare the drawing themselves? If so, it launches the link to the appropriate program. If not, permission for reprinting is lacking.

– Line 138 – Can a material belonging to the mesoporous group exhibit type I isotherm?

– Line 182 – “blue-shift” ­– this term is obsolete and I do not recommend its use to describe the IR spectra presented in the wavenumber scale. It would be better: “shift towards lower wavenumbers”.

– The structural characteristics of materials before and after irradiation (XRD, SEM) should be completed

– Authors also should clearly indicate what is innovative in this work.

The above observations do not allow the reviewer to recommend the paper in its current form for publication. I believe that the work requires additional research and analysis in order to be complete for publication.

Reviewer 2 Report

Presented paper titled: Photoswitchable zirconium MOF for light-driven hydrogen storage present an interesting approach to modification of UiO-66 MOF and as such could be used as a hydrogen storage.

In my opinion, the presented experiment was carried out very well. The collected data allowed for the assessment of the structure and microstructure of the tested material. There remains a certain insufficiency of spectroscopic studies, because in the light of such subtle changes it seems reasonable to use another method, e.g. Raman spectroscopy. It would allow to confirm the observations made with  FT-IR spectroscopy.

In my opinion, presented  work is correctly written and is suitable for publication after minor linguistic corrections and addition of suggested structural research.

Reviewer 3 Report

Butova and colleagues describe the modification of UiO-66-NH2 with diarylethene molecules. This material has been employed as a photoresponsive hydrogen container.

I think that the work is too preliminary and should not be published in Polymers.

There are some techniques that the authors must use in order to characterize their system: TGA, Elemental Analysis, solid-state NMR and SEM.

The functionalization percentage should be measured. I suggest to the authors an acid digestion of the material, analyzing the free ligands by NMR.

In addition, the authors should test their hydrogen adsorption experiments using additional wavelengths. I suggest experiments using 254 nm and 312 nm lamps.

It would also be interesting to perform XRD analyses before and after irradiation and to carry out an iterability study of the system.